# Metabolic Fingerprinting of Muscat of Alexandria Grape Musts during Industrial Alcoholic Fermentation Using HS-SPME and Liquid Injection with TMS Derivatization GC-MS Methods

**DOI:** 10.3390/molecules28124653

**Published:** 2023-06-08

**Authors:** Maria Marinaki, Thomai Mouskeftara, Panagiotis Arapitsas, Kyriaki G. Zinoviadou, Georgios Theodoridis

**Affiliations:** 1Laboratory of Analytical Chemistry, School of Chemistry, Aristotle University of Thessaloniki, 54124 Thessaloniki, Greece; 2BiomicAUTh, Center for Interdisciplinary Research and Innovation (CIRI-AUTH), 57001 Thessaloniki, Greece; 3FoodOmicsGR Research Infrastructure, AUTh Node, Center for Interdisciplinary Research and Innovation (CIRI-AUTH), 57001 Thessaloniki, Greece; 4Department of Medicine, Aristotle University, 54124 Thessaloniki, Greece; 5Department of Wine, Vine and Beverage Sciences, School of Food Science, University of West Attica, 12243 Athens, Greece; 6Research and Innovation Centre, Fondazione Edmund Mach, 38010 Trento, Italy; 7American Farm School, Perrotis College, 57001 Thessaloniki, Greece; kzinov@afs.edu.gr

**Keywords:** grape must, Muscat of Alexandria, wine metabolomics, volatile compounds, HS-SPME, derivatization, GC-MS

## Abstract

Muscat of Alexandria is one of the most aromatic grape cultivars, with a characteristic floral and fruity aroma, producing popular appellation of origin wines. The winemaking process is a critical factor contributing to the quality of the final product, so the aim of this work was to study metabolomic changes during the fermentation of grape musts at the industrial level from 11 tanks, 2 vintages, and 3 wineries of Limnos Island. A Headspace Solid-Phase Microextraction (HS-SPME) and a liquid injection with Trimethylsilyl (TMS) derivatization Gas Chromatography-Mass Spectrometry (GC-MS) methods were applied for the profiling of the main volatile and non-volatile polar metabolites originating from grapes or produced during winemaking, resulting in the identification of 109 and 69 metabolites, respectively. Multivariate statistical analysis models revealed the differentiation between the four examined time points during fermentation, and the most statistically significant metabolites were investigated by biomarker assessment, while their trends were presented with boxplots. Whilst the majority of compounds (ethyl esters, alcohols, acids, aldehydes, sugar alcohols) showed an upward trend, fermentable sugars, amino acids, and C6-compounds were decreased. Terpenes presented stable behavior, with the exception of terpenols, which were increased at the beginning and were then decreased after the 5th day of fermentation.

## 1. Introduction

“Muscat of Alexandria” is a variety of grape that has been cultivated since ancient times and is used not only for the production of table grapes and raisins but also for the production of white wines with a floral and fruity aroma. The largest percentage of its production goes towards producing dry, aromatic white wines and a few sparkling, fortified, famous dessert wines with appellation of origin and concentrated musts. In addition, due to the intense aromatic profile, this variety is also used in the vinification of white multi-varietal wines to improve the final aromatic composition of white wines [1,2]. Limnos Island is one of the predominant Greek locations producing sweet and dry white wines exclusively from Muscat of Alexandria grapes. The volcanic soil of Limnos supplements the variety’s attributes to their utmost, resulting in the production of two appellations of origin wines: the sweet Protected Designation of Origin (PDO) Muscat of Limnos wine and the dry PDO Limnos wine [3,4].

Volatile aroma is of primary interest in the evaluation of a wine’s quality, particularly with regards to white Muscat cultivars [5]. The aroma of wines is influenced by several factors: cultivar, region, climate, fermentation and yeast strain, vinification process, and storage conditions [6]. The typical “bouquet” of wine is a complex combination of hundreds of components belonging to different chemical groups [7], some of which are derived from grapes and others formed during alcoholic fermentation.

Varietal aroma is an important characteristic in wine sensorial characteristics and quality [8]. Wine yeasts are able to release varietal aroma compounds, consisting of terpenes, C13-norisoprenoids, and benzoic derivatives [9]. Terpenes is a characteristic group of wine aroma, especially for Muscat of Alexandria wines, possessing to primary wine aroma, as they are found mostly in grape skins and are transferred to wine during maceration [10,11,12]. In fact, these compounds are closely associated with the sensory expression of a wine’s bouquet, contributing to flowery odors, which are used for variety characterization [5,13,14]. Terpenols, such as linalool, nerol, geraniol, and *alpha*-terpineol, except their free volatile form and can be present as non-volatile, glycosydically bound precursors, and their enzymatic hydrolysis during fermentation can result in an improvement in wine aroma [15,16,17,18]. Benzene derivatives (benzyl alcohol and 2-phenylethanol), together with C6-alcohols (hexanols and hexenols, responsible for herbaceous character), are also volatile compounds originating from grapes, detected in grape musts and wines [2].

The secondary aroma of fermented beverages that affect a wine’s flavor and quality consist of various groups of compounds [1,19,20]. Higher alcohols, acids, ethyl and acetate esters, and carbonyl compounds are quantitatively dominant in secondary aroma, especially in quality criteria of white wine [21]. Ethyl esters are the main compound group of secondary metabolites in white wines, produced by yeasts during alcoholic fermentation from the esterification of fatty acids with ethanol, imparting pleasant, fruity, and floral characteristics [1,3,4,5], while the corresponding short chain fatty acids display odors described as cheese, sweat, rancid, or acid [1,2,3].

The non-volatile composition of wine is characterized, mainly, by the composition of a complex mixture of compounds at various concentrations. In grapes, the most quantitatively important compounds are two sugars, fructose and glucose. Disaccharides, such as sucrose or trehalose, non-fermentable sugars (xylose, mannose, arabinose), sugar alcohols (mannitol, sorbitol, arabitol), sugar acids, and glucosides complement the composition of grape musts in carbohydrates. Glycerol is the most abundant sugar alcohol in grape must, while inositols are observed in Muscat of Alexandria musts mainly with the major (*myo*-) and one of the minor (*scyllo*-) forms [11,20]. Organic acids are important for wine stability; they contribute to the organoleptic characteristics (flavor, color) and are strongly connected to the aroma of wine [22]. Some small organic acids are widely considered as grape-related organic markers (e.g., tartaric and malic acid) and alcoholic fermentation-related markers (e.g., caffeic and citric acid) [23].

Several researchers have studied the volatile profile of Muscat of Alexandria wines using various extraction methods and detectors [17,24]. Hardy [25] presented a Gas Chromatography coupled with Flame Ionization Detector (GC-FID) method, after Liquid–Liquid Extraction (LLE) of samples in order to study the volatilome of Muscat of Alexandria grapes during ripening, while a GC-FID method was also performed by van Rensburg et al. [26] to study the effect of a polysaccharide-degrading wine strain of *Saccharomyces cerevisiae* to the volatile metabolic profile of Muscat wines. Gas Chromatography coupled with Mass Spectrometry (GC-MS) is the first choice for the analysis of volatile compounds due to its sensitivity, efficiency, and reproducibility. Solid Phase Extraction (SPE) [27] and Stir-bar Sorptive Extraction (SBSE) [28], followed by GC-MS, have been used in the past; however, Headspace Solid-Phase Microextraction Gas Chromatography-Mass Spectrometry (HS-SPME-GC-MS) is one of the most commonly used techniques for the determination of volatile compounds in Muscat of Alexandria grapes, musts, and wines [5,29,30,31].

GC-MS, is not only the first choice for the volatile analysis but, along with Liquid Chromatography-Mass Spectrometry (LC-MS) [32,33,34], is commonly used for the profiling or for the quantification of polar non-volatile metabolites in a variety of substrates. One major advantage of GC-MS is that it has a relatively comprehensive range of coverage, thus permitting the analysis of organic and amino acids, sugars, and sugar alcohols in various samples [35]. However, derivatization is an additional step of sample preparation, required to convert the polar/semi-polar metabolites into their volatile counterparts, to facilitate analysis by GC-MS. Many derivatization methods are available, with alkylation and silylation being the most widely used protocols in metabolomics analysis and metabolic profiling. Silylation is a classic derivatization method, which is safe, easy to use, and able to derivatize sugars and sugar alcohols in contrast to alkylation reactions. 2,3,4,5,6-pentafluorbenzylbromide (PFBBr) [36] and tiethylamine (TEA) with ethyl acetate (EtAc) [37] have been used as reagents of derivatization for the identification of hydroxy-acids and biogenic amines; however, *N*,*O*-Bis-(trimethylsilyl)-trifluoroacetamide (BSTFA) [23,38,39] and *N*-Trimethylsilyl-*N*-methyl trifluoroacetamide (MSTFA) [35,40,41] with trimethylsilyl (TMS) are the most popular reagent for the derivatization of Muscat of Alexandria samples, as they offer a wide range of identified metabolites.

The aim of this work was to study the volatile and the polar non-volatile profiling of Muscat of Alexandria musts during alcoholic fermentation. To the best of the authors’ knowledge, few researchers have studied the volatile composition of white musts or the final product [2,9,42] or the fermentation of red wine [43]. This is the first time that a HS-SPME-GC-MS method and a liquid injection GC-MS with derivatization method have been applied in the analysis of Muscat of Alexandria musts during fermentation to monitor how the metabolome of musts changes throughout this complicated vinification process. Multivariate statistical analysis, Principal Components Analysis (PCA), and Orthogonal Projection to Latent Structures Discriminant Analysis (OPLS-DA) plots were also performed to distinguish the most statistically significant metabolites responsible for the differences between the stages of alcoholic fermentation.

## 2. Results and Discussion

In order to fulfill the aim of the project, to study the volatile and the polar non-volatile profiling of Muscat of Alexandria musts from Limnos Island during alcoholic fermentation, 44 samples, originated from 11 industrial scale fermentations, were analyzed. The idea was to cover a wide biological range of Limnos Island Muscat of Alexandria wine production, so for the first year of the project samples were collected from three wineries (3 industrial scale fermentations per winery), which gather 90% of the total production of dry white wine of the island. In the second year, samples were collected from two industrial scale fermentations only from the major winery that produces 70% of the total quantity of this wine in Limnos, according to data provided by the wineries and the competent service of the Municipality of Limnos Island. The samples were collected on the 1st, 5th, 8th, and 13th day of alcoholic fermentation by each of the 11 tanks, and in the following paragraphs are presented the outputs of the analysis. Our main goal was to investigate the behavior of a wide range of metabolites during fermentation of the aromatic Muscat of Alexandria must by having a broad biological variability, and not to find the differences between the wineries or vintages.

### 2.1. Volatile Composition of Grape Must—Changes during Fermentation

The composition of the volatile compounds of Muscat of Alexandria musts can be grouped into two major categories: the varietal volatile components (primary aroma) and the volatiles produced during alcoholic fermentation (secondary aroma). The special aromatic character of Muscat of Alexandria wines is attributed to the presence of terpenes and C13-norisoprenoids in concentrations higher than their odor threshold; such volatile molecules originate from grapes [44]. The major fermentation compounds, produced due to the metabolic activity of the yeasts, are alcohols, ethyl esters, acetates, and fatty acids [45]. Many primary and secondary volatile metabolites have been detected, identified, and semi-quantified with the described HS-SPME-GC-MS method, and a characteristic chromatogram of the 1st day of fermentation is shown in Figure 1. The volatile compounds found in Muscat of Alexandria musts, their relative concentrations on the 1st, 5th, 8th, and 13th day of alcoholic fermentation, and their experimental and bibliographic retention indices are presented in Appendix A.

In Muscat of Alexandria cultivars, the content of volatile aromatic compounds of the grape is inextricably linked to the quality of the produced wine, mostly depending on the terpene content and profile. Linalool, *alpha*-terpineol, ho-trienol, geraniol, and nerol are the most abundant oxygenated monoterpenes in Muscat of Alexandria musts at the 1st day of fermentation, and this is also observed for other “Muscat” musts and grapes [44,46,47,48]. In particular, linalool and *alpha*-terpineol, obtained by the metabolism of mevalonic acid, are responsible for the typical floral aromas of Muscat wines [47,49,50]. These compounds, known as varietal markers with aromatic character of sweet, rose-like, flowery notes, are almost unchanged, with small increases or decreases during fermentation, which is in agreement with earlier studies on Muscat of Alexandria varieties. 

Lanaridis et al. [2] studied both the free and the glycosidically linked forms of linalool, geraniol, and nerol in Muscat of Alexandria of Limnos Island grapes and wines; they observed a small decrease in these oxygenated monoterpenes in the transition of grape to wine. The glycoside forms of terpenes do not have a direct contribution to wine aroma, but they are quantitatively important as they represent the grape aromatic potential, because they are hydrolyzed to free volatile molecules during fermentation, both by the enzymatic action of yeast and by the acidic conditions; this could be a reason for the increase in some terpenes in this study, such as linalool and *alpha*-terpineol by the 5th day or nerol and *trans*-2-pinanol during the whole fermentation [16,18,46]. It is worth noting that several researchers have found that the percentage of glycosidically linked terpenes is higher than the free terpenes in Muscat grape cultivars, including the Muscat of Alexandria [15,51]. Whilst linalool and geraniol are the major aromatic terpenes, they are unstable in CO_2_ environments, so they are oxidized in small percentages to form more stable forms or isomers, as previously reported by Marais [52]. This fact could explain their decrease after the 5th day of fermentation (oxygenation process is present in all analyzed tanks at the first days of fermentation) and the increase in other more stable terpenols. 

Other terpenes (monocarbohydrates), such as D-limonene, *beta*-myrcene, *o*-cymene, *gamma*-terpinene, and 4-carene, present great stability during fermentation (Figure 2), with the first two being predominant, as also reported by Hardy [25], and the same behavior is also displayed by the detected C-13 norisoprenoids (*alpha*- and *beta*-ionone, *beta*-damascenone), originating from the direct degradation of grape carotenoids, which complement the primary aroma of the Muscat of Alexandria cultivar [46]. Boxplots of representative metabolites during fermentation of the most important groups of wine volatilomes are presented in Figure 2.

The ethyl esters, as for example ethyl butyrate, hexanoate, heptanoate, octanoate, nonanoate, and decanoate, sharply increase from the 1st to the 5th day of alcoholic fermentation of Muscat of Alexandria grape must and then exhibit a milder increase, stabilization, or even a decrease in their concentration, as illustrated in Figure 2. Similar results for the change of ethyl esters during alcoholic fermentation have previously been presented by Zhang et al. [43], and these observed an increase in these ethyl esters during Syrah wine fermentation when ethanol increased from 2% to 4%, and then a slight decrease by the end of the fermentation. On the other hand, Soares et al. [12] reported that ethyl esters increased throughout the fermentation of Moscatel sparkling wines, a fact that is in accordance with the results of this study for long chain fatty acid ethyl esters (ethyl dodecanoate, hexadecanoate, and 9-hexadecanoate). Acetates, formed during alcoholic fermentation from acetic acid and higher alcohols by enzymatic acetylation, as amyl, isoamyl, isobutyl, and phenylethyl acetate, increase during fermentation, presenting the highest concentration at the last day of fermentation, giving wines characteristic fruity notes [53]. Hexyl and octyl acetates are two exceptions, because their concentration highly increases at the beginning of fermentation and then presents a small decrease in the last few days. Yeast strains and the concentration of higher alcohols mainly affect the formation of acetate esters, while the fermentation conditions (rate, temperature, and oxygen exposure) could determine the formation of ethyl esters [54].

Another group of secondary aroma compounds that are important for wine chemistry due to their stability in the wine environment are the higher alcohols as they do not react strongly and participate in esterification and oxidation. This leads to the formation of esters and aldehydes, respectively [11]. The majority of higher alcohols are yeast by-products, as 2-methyl-1-propanol and 2-methyl-1-butanol; these molecules do not have the desired odor and produce aromas of petroleum, malt, and solvents, and are found in very small concentrations in wine. 2-Phenylethanol and isoamyl alcohol are by far the most abundant higher alcohols, derived from amino acid metabolism, through the catabolism of phenylalanine and leucine, respectively, and are therefore affected by the amount of nitrogen in musts [55]. Both 2-phenylethanol and isoamyl alcohol have been observed in high concentrations in both must and final product, imparting a pink and herbaceous/alcohol character at low concentrations, and rose and honey aromas at higher concentrations [10,12,56,57].

De Lorenzis et al. [58] presented an interesting observation, that the ratio between benzene derivatives (2-phenylethanol and benzyl alcohol) and oxygenated monoterpenes in Muscat of Alexandria berries is 5:95. However, this is not confirmed in this study due to the high concentration level of 2-phenylethanol. Conversely, benzyl alcohol or C6-alcohols, such as 1-hexanol and 3-hexen-1-ol, that were present in must, significantly decreased or disappeared after alcoholic fermentation. These compounds are known as pre-fermentative volatiles, are derived from grape fatty acids oxidation, and infer an herbaceous character to wines. After grape crushing, lipoxygenase catalyzes the transformation of lipids to alcohols, so that their concentrations increase during grape maceration [59]. The reduction of 1-hexanol content during winemaking may occur due to the formation of hexyl acetate, which highly increase at the first stages of winemaking [10,20].

Fatty acids, such as butanoic, hexanoic, octanoic, decanoic, and dodecanoic acid, have also been identified in Muscat of Alexandria musts. Octanoic acid, followed by decanoic acid, was found in musts in 10-fold higher concentrations than the rest of the acids and even shows a significant increase during the first stages of fermentation. These fatty acids are believed to be produced by yeasts as intermediates in the biosynthesis of long-chain fatty acids, starting with the formation of acetyl coenzyme A (acetyl-CoA) from the oxidative decarboxylation of pyruvic acid, followed by the action of the fatty acid synthase complex [54]. It is one of the most abundant metabolites, in accordance with its corresponding ethyl ester, both at the beginning and at the end of fermentation. All fatty acids, offering rancid, waxy, and cheesy odor perceptions, present the same trend as octanoic acid during fermentation, with a peak at the 5th day and a plateau or a small decrease until the end of fermentation, as can be seen from Figure 2 and Figure 3.

Whilst 3-methylbutanal, hexanal, and benzaldehyde [1,27,60] are the most abundant carbonyl compounds in Muscat of Alexandria musts, they present differing behaviors during fermentation. While 3-methylbutanal increases, benzaldehyde decreases, following the trend of their corresponding alcohols. Stevens et al. [44] and Webb and Kepner [61] detected C6-aldehydes, such as hexanal and 2-hexenal, in Muscat of Alexandria oils and wines, respectively, but only the first one has been detected in this study. 1-Ethoxy-1-methoxyethane, 1,1-diethoxy-3-methylbutane, and pentane-2,3-dione, which have been detected in very low concentrations in Muscat of Alexandria musts, have also been identified by Stevens et al. [44] in Muscat of Alexandria oil.

### 2.2. Metabolic Profiling of Sugars and Organic Acids during Fermentation 

Methoximation and silylation are the two steps of the applied derivatization process. Methoximation is necessary for preventing ring formation in the case of sugars and sugar derivatives in order to reduce the number of stereoisomers; methoximation is followed by the replacement of hydrogens by TMS moiety. This derivatization procedure has been proposed by several researchers for the identification of metabolites such as sugars and amino and organic acids, which are involved in many metabolic pathways in various matrices (food, wine, urine, blood) [35,40,62,63]. The stage of silylation is important as it is affected by various factors, such as the reaction temperature, and can lead to multiple peaks of the same compound in the case of incomplete silylation, as is observed in this study for fructose and glucose, which are the most abundant metabolites in grape musts, or in the study of Villas-Bôas et al. [40] for most of the detected sugars.

Grape must is largely composed of sugars (~about 200 g L^−1^ at the beginning of fermentation), making it difficult to detect less concentrated metabolites. High-throughput analysis and accurate identifications of hundreds of metabolites in a single analysis are just some of the factors that make GC–MS a very good choice for mapping the profile of musts during alcoholic fermentation. However, unstable or not sufficiently volatile or non-derivatized metabolites are not amenable to analysis by GC-MS. With the proposed method, 69 metabolites were identified, including mainly sugars and sugar alcohols, organic acids, and four amino acids; a chromatogram of a 1st day of fermentation grape must sample is presented in Figure 4.

This method aimed to study the general trends of the identified compounds, because sugars and sugar derivatives and amino and organic acids participate in many metabolic pathways during fermentation, and it is of particular interest to study them in an aromatic variety such as the Muscat of Alexandria cultivar. It is well known that grape must composition varies due to the differences in climatic conditions, such as water deficit, excess rainfall, or exposure to sunshine, mechanisms that are complex and related to physiology and the interaction with the environment of grapevines, e.g., the dry climate and volcanic soil on Limnos Island. Thus, the obtained data were correlated to the analyte peak areas of the 1st day of alcoholic fermentation; hence, the relative changes at the 5th, 8th, and 13th day in relation to the 1st day were expressed as ratios of peak areas (fold change). The identified metabolites, their retention time, chemical formula, and fold change between the stages of fermentation for each metabolite are presented in Appendix A.

Glycolysis, alcoholic fermentation, amino acid metabolism, and TCA cycle are the main metabolic pathways in which sugars and organic and amino acids participate [64]. Glucose and fructose are the main sugars that are converted into ethanol by the fermentation yeasts [65]. *Saccharomyces cerevisiae* seems to be glucose-friendly, and this is the reason why glucose is consumed faster than fructose. The consumption of these sugars by non-Saccharomyces yeasts has been studied by Miranda et al. [66] using LC-MS analysis of Madeira wine samples. Other pentoses (arabinose, lyxose, xylose, ribose) or hexoses (mannose, allose, *alpha-*, *beta-* mannose) are partially converted or not converted in ethanol; finally, some disaccharides (cellobiose, mannobiose, gentibiose) are not converted at all. An exception is sucrose, which, although it is not metabolized by yeasts, it is hydrolyzed by enzymes to its components, glucose and fructose, allowing it to be further metabolized. Pentoses remain in wines in small amounts, but the sweetness of wines is determined by the remaining glucose and fructose [64,67,68,69]. A small decrease in hexoses and pentoses (mannose, arabinose, ribose) is observed during fermentation, and this may be caused by their reaction with ethanol for the production of the sugar alcohols (arabitol, mannitol, sorbitol, ribitol). A decrease in mannose has also been reported by Balmaseda et al. [70], and it is strongly associated with the concentration of malic acid during and after malolactic fermentation. This trend of arabinose and ribose has also been observed at the degradation of Syrah grape biomass [71,72]. However, in this study, a release of non-fermentable sugars (mannose, arabinose, xylose) is observed, which could arise from peptic polysaccharides or glycoproteins, as has previously been reported by other researchers [73,74,75,76,77,78]. Different sugars and sugar alcohol metabolites have been previously identified by Cuadros-Inostroza et al. [79]; these were studied during the ripening process of Merlot and Cabernet grapes, again using TMS derivatization. The boxplots showing the trends of representative differentiating metabolites of the present study are illustrated in Figure 5.

TCA cycle and fatty acid metabolism of yeasts is probably related to the activity of the Ehrlich pathway for amino acid degradation. Malic and tartaric acid are the main organic acids found in grape juices at high concentrations, while succinic, citric, citramalic, caffeic, glutaric, and gluconic acids are present in musts in very small amounts. Degu et al. [80] studied the fold change of these organic acids during ripening, before veraison until maturity, but not during alcoholic fermentation, confirming the abundance of malic and tartaric acid at maturity against the rest of the detected acids. They presented a decreasing trend during fermentation; however, Walker et al. [81] observed a small increase in the first days of fermentation and then a final decrease in Muscat and Syrah wines. The rest of the organic acids presented an increase during alcoholic fermentation. These organic acids were only known for their contribution to the organoleptic properties and acidity of wines. However, Pinu et al. [41] observed that some of these acids, specifically citric and citramalic, negatively affect the production of varietal thiols [82].

Amino acids are consumed by yeasts as a source of nitrogen at the beginning of alcoholic fermentation, so this could be a reason for the detection of only four amino acids in Muscat of Alexandria musts. Among these four amino acids, only serine was consumed quickly and to a large extent; glycine was consumed to a lesser extent, GABA, an amino acid derivative, showed very little change, while proline showed a small increase. These findings are consistent with the study by Pinu et al. [83], who separated amino acids based on their rate of consumption by yeasts, with serine being in the highest consumption group and proline and GABA being not consumed at all. The explanation of the upward trend of proline could be that yeasts do not use proline as a nitrogen source. Therefore, proline is not consumed, and in addition it is released by the hydrolysis of oligopeptides during fermentation.

A map of the correlations between the identified metabolites is shown in Appendix A. It is obvious that the majority of metabolites present good correlation, because of their upward trend; however, it is observed that glucose and fructose are correlated only with each other, as they are the only metabolites consumed at such a very rapid rate during fermentation. Lactones highly increase at the beginning of fermentation and then decrease, presenting a unique trend that is not easily correlative, and some sugars (ribose, mannose, allose, galactose) also present a consuming trend or they remain stable, which is relative or absolutely opposite the general trend of the rest of the metabolites during alcoholic fermentation. 2,3-Butanediol and 2-phenylethanol are the only two metabolites detected by both methods; they exhibited a highly increasing trend (Appendix A), especially 2-phenylethanol, which is one of the most abundant compounds at the end of fermentation, presenting the exactly opposite trend with the fermentative sugars (Appendix A).

### 2.3. Multivariate Statistical Analysis during Alcoholic Fermentation for the Proposed Methods

For both methods, HS-SPME-GC-MS and liquid injection GC-MS after derivatization, all samples were analyzed in a single run, in random order. Along with the test samples, QC samples were analyzed in order to estimate analytical repeatability and system stability. Six QC samples were analyzed for the SPME method; eleven QC samples were analyzed in the liquid injection sample set. In both cases. the relative standard deviation of the peak areas of the identified metabolites was found to be below thirty percent (RSD < 30%). Multivariate statistical analysis was used to reveal differences and metabolites showing statistically significant differentiation between the stages of alcoholic fermentation.

PCA and PLS-DA score plots are illustrated in Figure 6. It is clear that the samples of the first day of fermentation are differentiated from the samples of the three other time points in both methods. This is in accordance with wine chemistry. In the case of volatiles, primary aromas from grapes are dominant at the beginning of fermentation; in the case of liquid injection analysis after derivatization, the sugars are in very high levels at the beginning of fermentation. As referred to in Appendix A, the numbers of samples express the tank and the date of their sampling, so a scattering of several samples of the same stage of fermentation was observed. For example, the samples 806, 1006, and 1106 in the PCA plot in Figure 6a are grouped closer to the second stage of fermentation, whilst they are 1st day’s must. These samples were supplied by the same winery, which produces certified Muscat of Alexandria white dry organic wines, made from organically grown grapes that do not contain any chemical fertilizers, pesticides, hormones, or genetically modified organisms, following all the guidelines of the legislation of Greece for organic wine production. Therefore, this differentiation could be retrieved from a different winemaking process or because of the production of these musts from organic grapes.

Moreover, an interesting observation from Figure 6 is that although the 1st and 5th day of fermentation are clearly separated in both methods, the 8th and the 13th day are difficult to discriminate, even in supervised PLS-DA plots. This can lead to the conclusion that the most important metabolic changes existed in the first days of alcoholic fermentation. In order to further study these differentiations, OPLS-DA models were applied. Appendix A show the OPLS-DA score plots from the models carried out between the 1st and the 13th day, 1st and 5th, 5th and 8th, and 8th and 13th day of fermentation, respectively, for both analyses. The statistical significance of these models was studied by CV-ANOVA analysis, permutation tests, and hoteling lines. For the HS-SPME method, the *p*-values for the first three models were *p1*^(1−13)^ = 1.02 × 10^−8^, *p2*^(1−5)^ = 0.00032, *p3*^(5−8)^ = 0.0062 and for the last one *p4*^(8−13)^ > 0.05. For the derivatization method, the corresponding values were *p1*^(1−13)^ = 3.76 × 10^−9^, *p2*^(1−5)^ = 0.028, *p3*^(5−8)^ = 1.19 × 10^−5^, and again the last one *p4*^(8−13)^ > 0.05. Appendix A show the significant metabolites between two stages of fermentation and their *p*-, p(corr) and VIP values as revealed by OPLS-DA models.

Twenty-two (22) metabolites were found to be statistically significant (*p* < 0.05, p(corr) > 0.5 and VIP > 0.8) for the 1st OPLS-DA model between the first and the last day of fermentation using the HS-SPME method. These included ethyl and acetate esters, terpenols, and benzene derivatives. Linalool, nerol, geraniol, ho-trienol, and *alpha*-terpineol were found among the significant metabolites; some of them were also the most abundant in musts. Ethyl octanoate, ethyl decanoate, octanoic acid, and 2-phenyethanol were found significant and were most abundant at the last day of fermentation. Twenty-three (23) and 18 metabolites were found to be significant for the next two models between 1st−5th and 5th−8th days of fermentation, while only eight metabolites were significant in the last model. Esters, alcohols, and acids were the main group of compounds responsible for the discrimination of the stages of fermentation. At the last two fermentation stages, only terpenols and fatty acids were found to be significant, while esters and alcohols were completely absent, with the exception of hexyl acetate.

In the models derived from liquid injection following derivatization, the number of significant metabolites was smaller compared to the models derived from HS-SPME-GC-MS data. The significant metabolites were10 for the model between the 1st and 13th day, 7 for the model between the 1st and 5th day, and 6 for the model between the 5th and 8th day, including mostly sugars as they are in very high concentrations in musts and overlap other metabolites. Fructose, glucose, galactose, mannose, and allose were among the most significant analytes for these three statistically significant models. For the OPLS-DA model between the last days of fermentation (8th and 13th), malic, tartaric, and succinic acids, as well as glycerol, along with sugars, seem to be among the eight most significant compounds. 

## 3. Materials and Methods

### 3.1. Chemicals

4-Methylpentan-2-ol was used as internal standard for the HS-SPME-GC-MS analysis; it was purchased from Merck (Darmstadt, Germany). *N*-alkanes (C8–C20) of analytical grade, Μethoxyamine hydrochloride, *N*-Methyl-*N*-(trimethylsilyl) trifluoroacetamide (MSTFA), Τrimethylchlorosilane (TMCS), pyridine anhydrous, and HCl 37% used for the derivatization of musts were obtained from Sigma-Aldrich (Darmstadt, Germany), while the sodium chloride used in the extraction of volatiles for the salting-out effect and the methanol (MeOH) used for the extraction before derivatization were purchased from Chem-Lab (Zedelgem, Belgium).

### 3.2. Samples

In total, 44 Muscat of Alexandria grape must samples were used for this study. All musts originated from Limnos Island in North Aegean Sea, from 3 different wineries in 2 different vintages (2019 and 2020). Musts were collected on the 1st, 5th, 8th, and 13th day of alcoholic fermentation, from 11 different tanks, and more specifically from 3 different tanks from each winery in 2019 and from 2 different tanks from 1 winery in 2020, produced in industrial conditions, with inoculation using commercial *Saccharomyces cerevisiae* yeasts. Appendix A presents additional information for the samples. Samples were stored in 2 mL aliquots at −50 °C. Quality Control (QC) samples were produced from equal portions of each sample.

### 3.3. HS-SPME-GC-MS Method

#### 3.3.1. Sample Preparation and Extraction

Sample extraction with HS-SPME and GC-MS conditions for the analysis of Muscat of Alexandria grape must samples were obtained from a previous protocol of Marinaki et al. [84] for the analysis of Greek PDO wines. Briefly, samples were thawed at room temperature and then 10 mL of each sample was placed in 15 mL centrifugation tubes and were centrifuged at 10,000 rpm for 15 min. In a 20 mL glass vial, 4.5 mL of the supernatant of centrifuged grape must and 13.5 μL of a 1000 μg mL^−1^ 4-methylpentan-2-ol internal standard solution (IS) in methanol were added, resulting in a final IS concentration of 3 μg mL^−1^. The vials were sealed with a polytetrafluoroethylene (PTFE)/silicone septum and were equilibrated at 50 °C for 7 min with agitation at 250 rpm using a PAL Shimadzu autosampler unit (AOC 6000, CTC Analytics, Zwingen, Switzerland). The Divinylbenzene/Carboxen/Polydimethylsiloxane (DVB/CAR/PDMS) fiber (Sigma-Aldrich, 2 cm length, 50/30 thickness) was then introduced to the headspace for 55 min at 50 °C. According to the manufacturer’s recommendations, the fiber was pre- and post-conditioned for 10 min at 260 °C. A QC sample was prepared as a representative sample by mixing equal volumes of each wine sample and was injected during the study to evaluate the stability of the analytical system. Blank runs were also performed to reveal possible carryover. The samples were analyzed in randomized order. QC samples were analyzed with the proposed method at the beginning, every 6 samples, and at the end of the batch of 44 samples.

#### 3.3.2. GC-MS Conditions

A Shimadzu GCMS-QP2020 instrument equipped with a PAL SHIMADZU autosampler unit (AOC 6000, CTC Analytics, Zwingen, Switzerland) was used for the analysis. Chromatographic separation was performed on a MEGA-5 MS capillary column (30 m × 0.25 mm, 0.25 μm) (MEGA, Legano, Italy). Briefly, the GC-MS conditions were as follows: Helium was used as carrier gas at a flow rate of 1.2 mL min^−1^; the column was held for 2.5 min at 40 °C, then programmed at 8 °C min^−1^ to 110 °C and at a rate of 10 °C min^−1^ to 240 °C and held for 5 min. Injection was operated in split mode (1:5) at 265 °C. The mass detector was operated in electron impact mode at 70 eV. The temperatures of MS source and quadrupole were set to 260 and 200 °C, respectively. The mass spectra scanned at *m*/*z* 35–450 amu range.

### 3.4. GC-MS Method using Liquid Injection after Derivatization 

#### 3.4.1. Sample Extraction and Derivatization

Aliquots of 2 mL of each sample were stored at −50 °C. Samples were thawed at room temperature and were then centrifuged at 4 °C and 14,000 rpm for 15 min. For the derivatization, a combination of 3 protocols [40,41,62] were used after a few trials of methoximation and silylation time and temperature incubation. An amount of 20 μL of the supernatant was mixed with 60 μL of methanol (MeOH), and the samples were evaporated to dryness under vacuum (Eppendorf Concentrator) at room temperature. The dried extracts were derivatized with 80 μL of methoxyamine (MeOX) 2% in pyridine, and the sample was incubated for 90 min at 50 °C. Silylation reaction was followed by the addition of 80 μL of MSTFA 1% TMCS and incubation for 60 min at 70 °C.

#### 3.4.2. GC-MS Conditions

Untargeted GC-MS/MS analysis was performed by an EVOQ 456 GC-TQMS (Bruker, Bremen, Germany), equipped with a CTC autosampler and a PTV injector, controlled by Compass Hystar software (https://www.bruker.com/en/products-and-solutions/mass-spectrometry/lc-ms/compass-hystar.html, accessed on 10 May 2023). Separation was performed on a 30 m HP-5 MS UI (Agilent J&W) capillary column (0.25 mm ID, 0.25 μm film thickness). The injection volume was 1 μL, and different split ratios (1:5, 1:10, 1:20) were tested. The oven program and the injector’s conditions are provided in detail in a previously published study [85].

### 3.5. Data Processing and Chemometrics

GC-MS data for both untargeted HS-SPME and liquid injection with derivatization methods were initially processed with AMDIS software (http://www.amdis.net/, accessed on 10 May 2023) for chromatographic peak deconvolution and identification. NIST and FIEHN libraries were used for the identification, applying simple mode with a minimum match factor of 70%. Peak areas of the compounds extracted by AMDIS were calculated using the Gavin3 script in MATLAB. The trend of derivatized metabolites during fermentation is expressed as the fold change of the peak area of each stage of fermentation (5th, 8th, and 13th day) compared with the 1st day. For volatile metabolite identification in the case of HS-SPME, the mass spectra of eluting compounds were compared to those of commercial library NIST14, using both AMDIS and Shimadzu software LabSolutions (https://www.shimadzu.com/an/products/software-informatics/index.html, accessed on 10 May 2023), GCMS Solutions version 2.50 SU3 Lab Solution, and linear retention indices were calculated relative to a series of *n*-alkanes (C8–C20) (Sigma-Aldrich). The volatile organic compounds (VOCs) were semi-quantified by dividing the peak areas of the compounds of interest by the peak area of the internal standard (4-methylpentan-2-ol) and multiplying this ratio by the initial concentration of the internal standard (expressed as mg L^−1^). In both cases, the peak areas were calculated from the full-scan chromatogram using total ion current (TIC). 

Multivariate statistical analysis and biomarker assessment via Variable Importance for the Projection (VIP) plots, loading plots, S-plots, p(corr), and Hotelling’s lines were performed using the Soft independent modelling by class analogy (SIMCA) package (version 14.1; Umetrics, Umeå, Sweden) and the online platform Metaboanalyst (https://www.metaboanalyst.ca/, accessed on 2 February 2023) [47,48]. PCA, PLS-DA, and OPLS-DA were performed to assess data in a multivariate setting. The validation of each model was evaluated using permutation plots and cross-validated Analysis of Variance (CV-ANOVA) values. Two-tailed *t*-test, with unequal variance and a threshold of *p* < 0.05, and ANOVA were employed in Microsoft Excel Spreadsheets.

## 4. Conclusions

This study presented the application of two complementary GC-MS protocols, which enabled the measurement of volatile (via HS-SPME) and non-volatile/semi-volatile (via liquid injection after derivatization) metabolites of Muscat of Alexandria grape musts during their industrial alcoholic fermentation. In total, 44 samples were analyzed, sampled from 11 tanks, 2 vintages (2019 and 2020), and 3 wineries of Limnos Island, by covering four time points of the alcoholic fermentation (1st, 5th, 8th, and 13th day). Through the HS-SPME-GC-MS protocol it was possible to follow the behavior of 109 metabolites (terpenes, terpenoids, alcohols, aldehydes, esters, norisoprenoids, etc.), while 69 metabolites (sugars, acids, amino acids, etc.) were followed by the complementary TMS derivatization GC-MS protocol. Chemometrics using PCA, OPLS-DA, and biomarker assessment led to the discrimination of four time points during fermentation and to reveal the most significant metabolites. 

Linalool, well known for being responsible for the characteristic aroma of Muscat of Alexandria wines, was found to be one of the most abundant volatile compounds, presenting the highest peak on the chromatogram and also presented an increase in the first days of fermentation and a decrease after the oxygenation process. A similar trend was observed for the majority of terpenols (*alpha*-terpineol, ho-trienol, nerol, and geraniol), which could decrease at the end of fermentation due to their oxidation in more stable forms. Primary aroma metabolites, such as terpenes (*beta*-myrcene, D-limonene) and C13-norisoprenoids (*beta*-damascenone, *beta*-ionone) showed a stable trend during fermentation, while C6-compounds (1-hexanol and hexanal) and benzene derivatives (benzyl alcohol and benzaldehyde) decreased. Moreover, ethyl and acetate esters, alcohols, aldehydes, and fatty acids are fermentation products and presented a sharp increase the first five days, followed by a milder increase or a stabilization towards the end of alcoholic fermentation. The most abundant of these groups of metabolites in Muscat of Alexandria grape musts were found to be octanoic acid and ethyl octanoate, 2-phenylethanol and isoamyl alcohol, and 3-methylbutanal, respectively. As expected, fermentable sugars and amino acids were catabolized, while most of the other non-volatile/semi-volatile metabolites analyzed after TMS derivatization showed an upward trend during fermentation. Glucose was consumed first by the 5th day of fermentation, as it is preferred by the yeasts, while fructose decreased in a milder way. 2-Phenylethanol and 2,3-butanediol presented the upward trend from the beginning of fermentation, while non-fermentable sugars and sugar alcohols were increased at the last stages of fermentation, due to the degradation of polysaccharides or the reaction with ethanol, respectively.

The outputs of this study provide valuable information for the winemakers and the wine scientists that work with the Muscat of Alexandria cultivar.

## Figures and Tables

**Figure 1 molecules-28-04653-f001:**
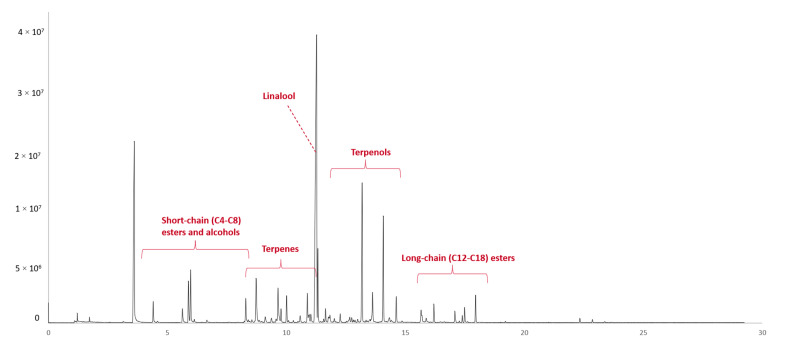
Representative HS-SPME-GC-MS chromatogram of volatile compounds identified in a 1st day Muscat of Alexandria grape must.

**Figure 2 molecules-28-04653-f002:**
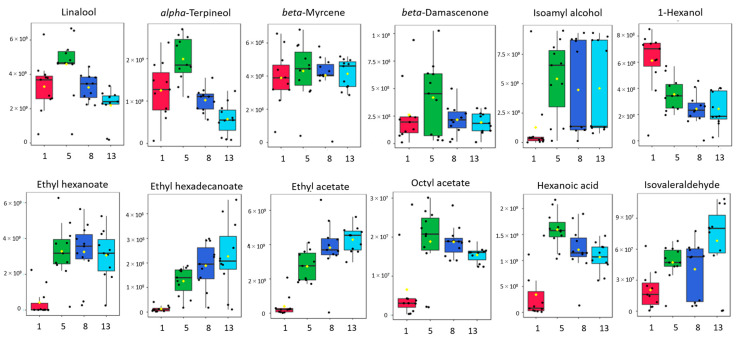
Boxplots illustrating the change of certain marker molecules during alcoholic fermentation: representative terpenes, C13-norisoprenoids, alcohols, ethyl and acetates esters, acids, and aldehydes.

**Figure 3 molecules-28-04653-f003:**
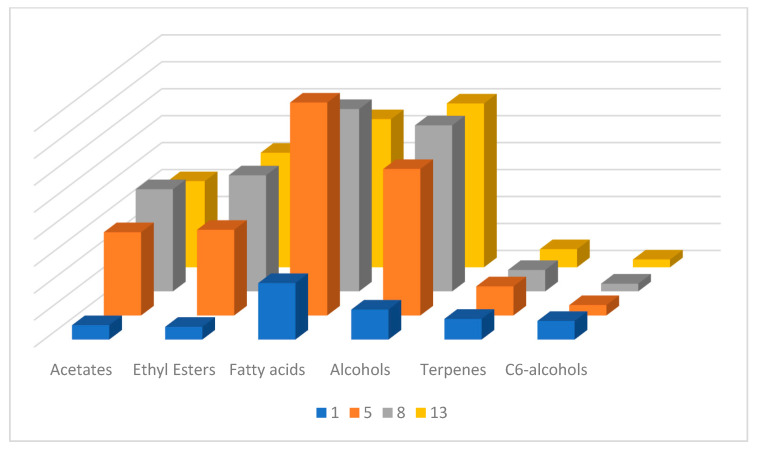
Trend of levels of major identified compound groups by HS-SPME-GC-MS analysis at four time points, 1st, 5th, 8th and 13th day of alcoholic fermentation.

**Figure 4 molecules-28-04653-f004:**
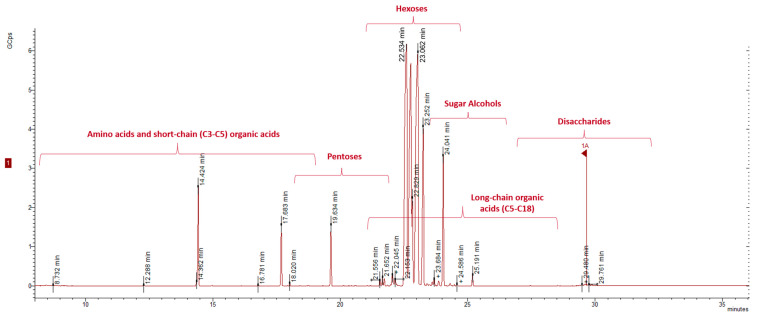
Representative chromatogram of GC-MS after derivatization of polar compounds identified in a 1st day Muscat of Alexandria grape must.

**Figure 5 molecules-28-04653-f005:**
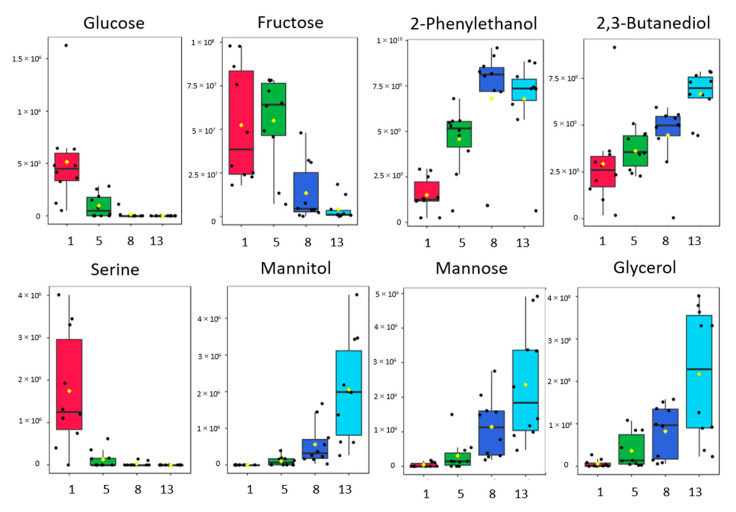
Boxplots showing the trends of representative compounds identified using the derivatization method (sugars, sugar alcohol, amino acids, alcohols, and acids) during alcoholic fermentation.

**Figure 6 molecules-28-04653-f006:**
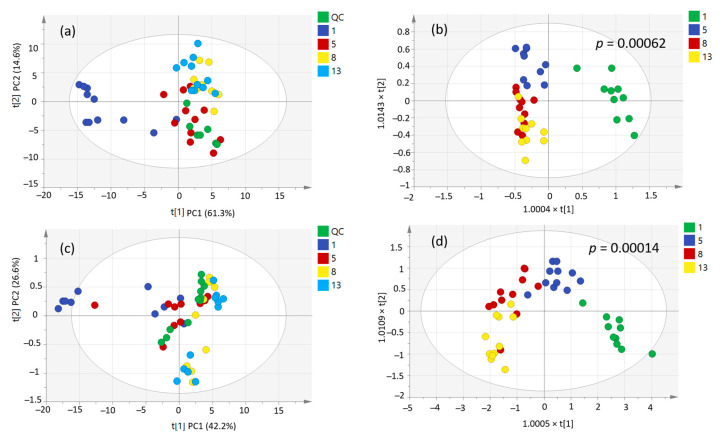
Scores plot showing the samples of days 1, 5, 8, and 13 of fermentation and the QC samples. (**a**,**b**) PCA and PLS-DA score plots of the HS-SPME-GC-MS method. (**c**,**d**) PCA and PLS-DA score plots of the liquid injection with derivatization GC-MS method.

## Data Availability

All data are included in the main text and the Appendix A.

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
