# Peer review of "Metabolic Fingerprinting of Muscat of Alexandria Grape Musts during Industrial Alcoholic Fermentation Using HS-SPME and Liquid Injection with TMS Derivatization GC-MS Methods"

_molecules, 2023, doi:10.3390/molecules28124653_

Round 1
Reviewer 1 Report
The aim of this research was to chemically characterize metabolomic changes during the fermentation of grape musts at industrial level of Limnos island by GC-MS with HS-SPME and TMS. It is an interesting manuscript, that could be eventually published if it will be extensively revised.
To improve the manuscript, I would suggest the following:
1) The fermentation process of wine was not described. It was natural fermentation or inoculation fermentation?
2) The total time of Tank 8 was 12, and how to take samples at day 12? The time of sampling (1, 5, 8, 13) was same?
The total time of fermentation was not same from Table S1, For tank 1, 2, and 3 was 16, 13, 13 respectively. Tank 8, 10, 11 was 14; others was 12, 13, 16, 11, and 12.
3) Samples were from 3 different wineries in 2 different years (2019 and 2020). The authors should explain the reasons. (why chose year of 2020, and chose 2 samples not 3 in a tank of 2020.) Whether all data was the average of all samples.
4) The grammar should be modified, eg. the past tense.
5) L99 It is not suitable.
Headspace Solid-Phase Microextraction Gas Chromatography-Mass Spectrometry (HS- SPME-GC-MS) is the state-of-the-art technique for the determination of volatile compounds in Muscat of Alexandria grapes, musts and wines.
6) L133 HS-SPME-GCMS should be changed to HS-SPME-GC-MS.
7) L359 “8th and 13th day was difficult to discriminate,” the authors should explain it.
8) L399 C should lower-case.
9) L353 The accumulative variance contribution rate of Figure 6 should be added.
10) In my opinion, conclusions should be rewritten; the source of major volatile compounds was not clear. The authors just described the changes during fermentation. Depth of discussion should be strengthened.
Eg. What were the main volatile compounds in wine of Alexandria grape? Only linalool?
11) Latin for strain should be italic.
12) The reference should be uniformed (like upper and lower case).
The grammar should be modified
Reviewer 2 Report
The article presented by the authors is well written, it has all the elements for its publication. For my part I did not find details or corrections to make.Author Response
Please see the attachment.

Reviewer 3 Report
Authors used two methods to have the metabolic fingerprinting of Muscat of Alexandria grape musts during industrial alcoholic fermentation. The results are very useful for wine makers and method could be used for similar processes too.
Comments:
1, Line 149-150, “Linalool, alpha-terpineol, hotrienol, geraniol and nerol are the most abundant Terpenes…”
Mostly Terpenes refer to (C5H8)n compounds. The mentioned compounds are oxygenated monoterpenes.
2, Linw 159, “…served a small decrease of these monoterpenes from grape to wine.”
Suggest changing monoterpenes to oxygenated monoterpenes. Monoterpenes are the C10H16 terpenes.
3, Wine is a mixture of many different chemicals. Did authors see matrix effect on the SPME method internal standard? If not, why? If yes, how did that matrix effect impact the result analysis (since it is not “similar” to most of the target compounds)?
4, Line 201-212, “higher alcohols”
What is the definition of “higher alcohols”? Isoamyl alcohol is not a “big” alcohol?
5, Line 214
Monoterpene, see comment 2.
6, Line 238-241 and line 241-246.
Suggest rewriting the sentences to make it more reader friendly. For example, “The most abundant carbonyl compounds in Muscat of Alexandria musts are 1-methylbutanal and benzaldehyde [1,27,61], which have completely opposite behavior during fermentation. While 1-methylbutanal increases, benzaldehyde decreases, following the trend of their respective alcohols.
7, Figure 4
It is impossible for C6-C9 organic acids to be separated by that much from C3-C5 organic acids.
8, Line 342-343, “especially 2-phenylethanol, which is of the most abundant compounds at the end of fermentation and …”
Since the concentration was “estimated” based on the assumption that target compounds had the same response factor as the internal standard had, to say 2-phenylethanol is the “most” abundant compounds is not an accurate statement. Should be changed to: one of the most abundant compounds.
9, Line 353-360
Rewrite the sentences to make them more readable. For example, “three other points” would be “three other time points”.
10, Line 366, “organic grape”
This can be misleading. People would consider “organic grape” as “grape produced organically”, such as non-GMO, no pesticide use etc. Authors want to say different “grape organics”? Rewrite the sentence.
11, Line 434-437. “A QC sample was prepared as a representative sample by mixing equal volumes of each wine sample and was injected during the study to evaluate stability of the analytical system. Blank runs were also performed to reveal possible carryover.”
If the QC sample was “injected”, how did the GC injector handle SPME and liquid injection without change liner? If authors want to say, run the QC sample between samples, should change the “injected” to “run” or “analyzed”.
12, Line 507. “Linalool found to be the major volatile compound, “
May consider to rewrite this statement.
13, In most of the Table Sx, and Figure S1, many compound names are listed as their chiral isomer names. That could be true in those samples. However, The GC columns used by the authors were not chiral columns. It is impossible for authors to claim a given peak is the chiral isomer listed. This has to be corrected.
14, In many of the Tables and Figure S1, some compounds have very “similar” names, for example, D-Fructose and D-(-)-Fructose; D-Glucose and d-Glucose. Author should at least explain what the meaning is.
Round 2
Reviewer 1 Report
The authors answered comments mainly. Please unify the reference, like upper and lower case, Latin for strain. The accumulative variance contribution rate of PCA should be added.
Author Response
We would like to thank the reviewer for the comments. We added the accumulative variance contribution rate of the PCA in Figure 6. We apologize for the references, we used Zotero to add or edit the citations and the bibliography of our manuscript and the reason of this inconvenience is that it adds the title of the reference as it is given from each journal. However, we edited the references (upper and lower case, Latin for strain) in order to be unified.